# Mortality and other outcomes of patients with coronavirus disease pneumonia admitted to the emergency department: A prospective observational Brazilian study

Rodrigo A. Brandão Neto[1¶], Julio F. Marchini[1¶]*, Lucas O. Marino[1], Julio C. G. Alencar[1], Felippe Lazar Neto[1], Sabrina Ribeiro[1], Fernando V. Salvetti[1], Hassan Rahhal[1], Luz Marina Gomez Gomez[1], Caue G. Bueno[1], Carine C. Faria[1], Victor P. da Cunha[1], Eduardo Padrão[1], Irineu T. Velasco[1], Heraldo Possolo de Souza[2], Emergencia USP Covid group¶

1 Emergency Department, Hospital das Clínicas da Faculdade de Medicina da Universidade de São Paulo, São Paulo, Brazil, 2 University of São Paulo Medical School, São Paulo, Brazil

¶ Complete membership of the author group can be found in the Acknowledgments.

* jfmarchini@gmail.com

**Data Availability Statement:** All data has been submitted here.

## Abstract

### Background

The first cases of coronavirus disease (COVID-19) in Brazil were diagnosed in February 2020. Our Emergency Department (ED) was designated as a COVID-19 exclusive service. We report our first 500 confirmed COVID-19 pneumonia patients.

### Methods

From 14 March to 16 May 2020, we enrolled all patients admitted to our ED that had a diagnosis of COVID-19 pneumonia. Infection was confirmed via nasopharyngeal swabs or tracheal aspirate PCR. The outcomes included hospital discharge, invasive mechanical ventilation, and in-hospital death, among others.

### Results

From 2219 patients received in the ED, we included 506 with confirmed COVID-19 pneumonia. We found that 333 patients were discharged home (65.9%), 153 died (30.2%), and 20 (3.9%) remained in the hospital. A total of 300 patients (59.3%) required ICU admission, and 227 (44.9%) needed invasive ventilation. The multivariate analysis found age, number of comorbidities, extension of ground glass opacities on chest CT and troponin with a direct relationship with all-cause mortality, whereas dysgeusia, use of angiotensin converting enzyme inhibitor or angiotensin-ii receptor blocker and number of lymphocytes with an inverse relationship with all-cause mortality

### Conclusions

This was a sample of severe patients with COVID-19, with 59.2% admitted to the ICU and 41.5% requiring mechanical ventilator support. We were able to ascertain the outcome in

**Funding:** Dr. Gomez was supported by FAPESP grant #2019/23078-1.

**Competing interests:** The authors have declared that no competing interests exist.

majority (96%) of patients. While the overall mortality was 30.2%, mortality for intubated patients was 55.9%. Multivariate analysis agreed with data found in other studies although the use of angiotensin converting enzyme inhibitor or angiotensin-ii receptor blocker as a protective factor could be promising but would need further studies.

## Trial registration

The study was registered in the Brazilian registry of clinical trials: RBR-5d4dj5.

## Introduction

In December 2019, several cases of pneumonia of unknown aetiology were reported in Wuhan, Hubei, China [1]. A few weeks later, a novel enveloped betacoronavirus named severe acute respiratory syndrome coronavirus 2 (SARS-CoV-2) [2] was identified as the etiologic agent, and the new disease was announced as coronavirus disease (COVID-19) [3]. Less than 6 months after its discovery, COVID-19 was responsible for more than 7 million cases and 400,000 deaths worldwide [4].

In Brazil, the first case of COVID-19 was diagnosed in February 2020, and the community transmission was established in March. In June, the World Health Organization (WHO) declared Brazil as the new epicentre of the disease, with more than 832,000 cases and 42,000 deaths [4].

São Paulo is the most developed and populous state in Brazil and is where the first COVID-19 cases were diagnosed in the country. Several temporary hospitals have been built following the surge of cases, and this service, the largest public hospital in the state, has been designated to provide medical care exclusively to patients with COVID-19.

In this study, we report our experience with these patients, all of whom were admitted owing to a confirmed or suggested diagnosis of COVID-19. Our objective is to report the characteristics of our patients, their clinical course during admission, their final outcome and finally to identify independent predictors of death. All of them had pneumonia of at least moderate severity, which has been considered as a defining characteristic of COVID-19 by the WHO [5], and they all required hospitalisation.

To our knowledge, this is the first report of a series of COVID-19 cases in the Southern Hemisphere. It is also noteworthy that, contrary to China, Europe, and the United States, the COVID-19 pandemic has affected Brazil during the warm weather. Comparing experiences among different countries and identifying changes in disease virulence and lethality over time may be vital for understanding and containing this pandemic.

## Methods

### Study design

From 14 March to 16 May 2020, all patients admitted to the Emergency Department (ED) of Hospital das Clínicas da Faculdade de Medicina da Universidade de São Paulo were enrolled in this study. The hospital is a 1000-bed quaternary academic medical centre and is affiliated to the University of São Paulo in the city of São Paulo, Brazil. During this pandemic, the state government designated this hospital to be the reference centre for all moderate and severe cases of COVID-19. Our ED admitted the vast majority of patients from the city. Moreover, patients suggested of having COVID-19 who were received in other institutions were also transferred to our ED. From these patients, this study analysed those who met the following criteria:

- Hospital admission longer than 6 hours

- Lung involvement, as diagnosed on chest X-ray or computed tomography (CT)

- Confirmed diagnosis of COVID-19

Diagnosis of SARS-CoV-2 infection was confirmed by performing real-time reverse transcription–polymerase chain reaction (RT-PCR) on nasopharyngeal swabs or tracheal aspirate specimens. The test was repeated in patients with negative RT-PCR results if the SARS-CoV-2 infection was still suggested. RT-PCR was performed in accordance with the Centers for Disease Control and Prevention (CDC) and WHO guidelines [5, 6]. The pneumonia diagnosis was confirmed via chest radiographs or lung CT. All pneumonia cases were confirmed by at least two ED physicians and a radiology expert.

A standardised form was used to collect demographic data from the patients as well as data related to the following variables: underlying medical conditions; medications used; clinical signs and symptoms at admission; laboratory tests; and outcomes, including discharge, in-hospital death, intensive care unit (ICU) admission, invasive mechanical ventilation, vasopressor use, renal failure, and renal replacement therapy. Additional information, including vital status during hospitalisation was updated through medical registers. A medical doctor, member of the research team, was responsible for cross-checking information and assuring quality control.

The primary end point was death, and the secondary end points were discharge, ICU admission, need for invasive ventilation, and vasopressor use in patients with COVID-19 pneumonia.

The study protocol was approved by the Research Ethics Committee (REC) of Hospital das Clínicas da Faculdade de Medicina da Universidade de São Paulo (protocol number CAAE 30417520.0.0000.0068) with written informed consent or verbal authorization documented in the patient's charts. Patient anonymity was preserved. Written informed consent was not possible, for example, if the patient was unconscious or in acute respiratory failure. When written informed consent from the patient was not possible the REC approved informed verbal authorization from the patient or family members in the presence of witnesses. All patients were treated according to hospital protocols, which included prescription of antibiotics at admission in all cases. The study was registered in the Brazilian registry of clinical trials: RBR-5d4dj5

### Statistical analysis

Descriptive statistics were calculated for all study variables. Data are expressed as absolute frequencies and percentages for categorical variables. For normally and non-normally distributed continuous variables, data are expressed as means and standard deviations and as medians with interquartile ranges, respectively. The patients' characteristics and outcomes were compared between survivors vs. non-survivors, ICU vs. non-ICU care and intubated vs. non-intubated. We tested univariate associations to the primary outcome with Student's t-test and the Kruskal–Wallis test for normally distributed and non-normally distributed continuous variables, respectively, whereas the chi-squared test was used to analyse categorical variables. First we applied the chained equations algorithm, that imputes incomplete multivariable data, As a result, five sets of imputed data were obtained [7, 8]. These datasets are stacked in a single matrix. A binomial family model with lasso was fitted [9]. Using the binomial family model with lasso we created a class of penalized objective functions which constructed a homogeneous pooled objective function. We combined the objective functions for each of the imputed datasets together and jointly optimized the collective objective function [9, 10]. All statistical tests were two-sided, and p-values $< 0.05$ were considered statistically significant. Study data

were collected and managed using REDCap electronic data capture tools hosted at this institution [11]. Statistical analyses were performed using StataCorp. 2013. Stata Statistical Software: Release 13. College Station, TX: StataCorp LP and using R version 4.0.3 (2020-10-10), packages miselect and mice.

### Definitions

Fever was defined as an axillary temperature of at least 37.8°C. Acute kidney injury was defined according to the clinical practice guidelines of the Kidney Disease Improving Global Outcomes (KDIGO) [12]. The illness severity of COVID-19 was defined according to the WHO guidelines [5]. Extension of pneumonia on CT was classified on an ordinal scale of 1–5 (1: <25%; 2: <50%; 3: 50%; 4: >50%; and 5: >75%), according to the official CT report. The number of comorbidities variable was created adding 1 for each of the following factors: hypertension, diabetes, smoking, hemodialysis, heart conditions (heart failure, atrial fibrillation or other heart diseases), lung conditions (COPD, asthma or lung diseases), cancer, cirrhosis or transplant patient. This variable was capped at 4.

## Results

A total of 2219 patients were received in the ED during the study period. Among them, 1193 stayed for at least 6 hours or were admitted to the ED. We confirmed SARS-CoV-2 infection in 518 patients and pneumonia in 506 patients who were enrolled in this study (Fig 1).

The demographic and clinical characteristics of the patients are shown in Table 1. Of the included patients, 57.3% were males, and the average age was 59.2 years (±16.8). Bilateral lung involvement was observed in all patients on chest X-ray or CT.

COVID-19, coronavirus disease; ICU, intensive care unit; COPD, chronic obstructive pulmonary disease; HIV, human immunodeficiency virus infection; ACEIs, angiotensin-converting enzyme inhibitors; ARBs, angiotensin II type 1 receptor blockers; NSAIDs, non-steroidal anti-inflammatory drugs

The patients have had the symptoms for 8.5 days on average, before they were admitted to the ED, and 394 (77.9%) patients had at least one comorbidity, of which the most prevalent was hypertension (found in 280 patients– 55.3%), followed by diabetes (found in 181 patients– 35.8%). Almost half of the patients had multiple comorbidities (249 patients– 49.2%). The most common symptoms were dyspnoea (385 patients– 76.1%), cough (376 patients– 74.3%), fever (231 patients– 45.7%), and myalgia (139 patients– 38.9%). In addition, 80 (15.8%) patients had rhinorrhoea and 75 (14.8%) had sore throat. Anosmia and dysgeusia, symptoms that were common in other series, were also present in 117 (23.1%) and 99 (19.6%) patients, respectively [13–15], while diarrhoea and nausea were present in 88 (17.4%) and 97 (19.2%).

Among the patients, 417 (82.5%) needed oxygen supplementation in the ED, 62 (12.4%) needed vasopressors immediately after arrival in the ED, and 44 (8.7%) were already intubated upon arrival to the ED.

Similar to other series, lymphopenia, defined as a lymphocyte count < 1500 cells/μL, was quite common 418 patients (82.8%) [5, 6]. D-dimer levels were elevated in 83.5% of patients. Levels of lactate dehydrogenase, C-reactive protein, cardiac troponin T, and creatine phosphokinase were also elevated in 446 patients (88.2%), 320 (63.4%), 193 (38.3%), and 138 (27.4%), respectively. Laboratory test results are shown in Table 2.

### Outcomes

We found that 153 patients died (30.2%), 333 were discharged home (65.9%), and 20 (3.9%) remained in the hospital when we finalised data collection on August 27, 2020 (Table 2). A

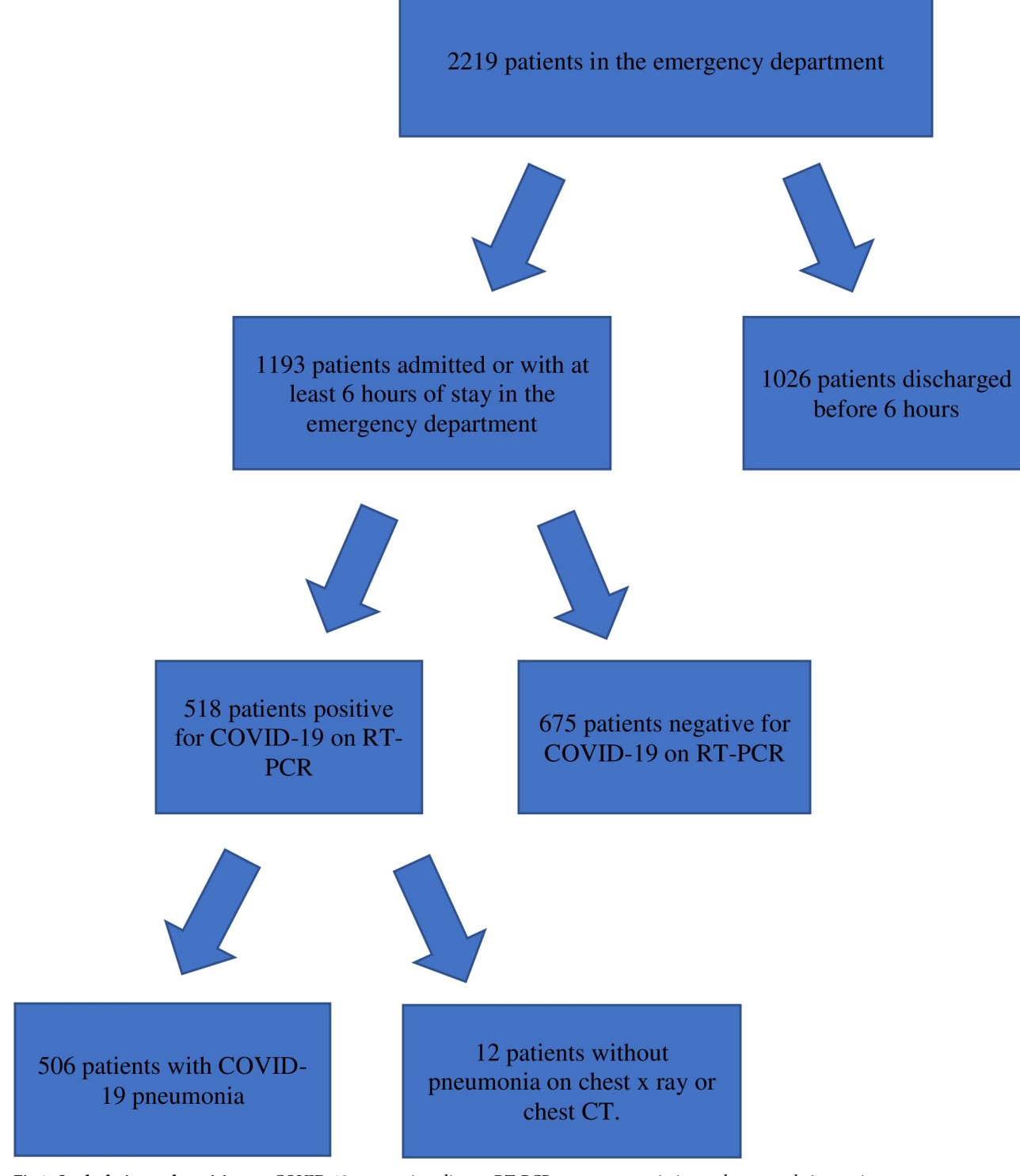

**Fig 1. Study design and participants.** COVID-19, coronavirus disease; RT-PCR, reverse transcription–polymerase chain reaction.

total of 300 of the 506 patients (59.3%) required ICU admission, 227 (44.9%) needed invasive ventilation, 179 (35.4%) needed vasopressors, and 68 (13.4%) required dialysis. Of the 353 survivors, 165 patients (46.7%) went through the ICU stay, 65 (18.4%) used vasopressors and 19

**Table 1. Characteristics of the 506 patients with COVID-19 pneumonia at admission to the emergency department.**

| Variable | All patients | Survivors | Non-survivors | P | Non-ICU patients | ICU patients | p | Non-intubated patients | Intubated patients | p |
|---|---|---|---|---|---|---|---|---|---|---|
| N | 506 | 353 | 153 | | 206 | 300 | | 279 | 227 | |
| Age, mean ± SD | 60.1±15.1 | 57.4±14.6 | 66.2±14.4 | <0.0001 | 58.4±15.5 | 61.2±14.7 | 0.0413 | 59.4±15.6 | 60.9±14.5 | 0.274 |
| Females, N (%) | 216 (42.7%) | 148 (41.9%) | 68 (44.4%) | 0.599 | 92 (44.7%) | 124 (41.3%) | 0.457 | 126 (45.2%) | 90 (39.7%) | 0.212 |
| White, N (%) | 268 (53.0%) | 186 (52.7%) | 82 (53.6%) | 0.852 | 164 (54.7%) | 104 (50.5%) | 0.355 | 142 (50.9%) | 126 (55.5%) | 0.301 |
| **Symptoms on presentation, N (%)** | | | | | | | | | | |
| Dyspnoea | 385 (76.1%) | 267 (75.6%) | 118 (77.1%) | 0.719 0.719 | 154 (74.8%) | 231 (77%) | 0.561 | 210 (75.3%) | 175 (77.1%) | 0.632 |
| Cough | 376 (74.3%) | 270 (76.5%) | 106 (69.3%) | 0.088 | 161 (78.2%) | 215 (71.7%) | 0.088 | 210 (75.3%) | 166 (73.1%) | 0.584 |
| Myalgia | 197 (38.9%) | 159 (45.0%) | 38 (24.8%) | <0.0001 | 98 (47.6%) | 99 (33%) | 0.001 | 127 (45.5%) | 70 (30.8%) | 0.001 |
| Sore throat | 75 (14.8%) | 60 (17%) | 15 (9.8%) | 0.036 | 38 (18.4%) | 37 (12.3%) | 0.057 | 48 (17.2%) | 27 (11.9%) | 0.095 |
| Rhinorrhoea | 80 (15.8%) | 56 (15.9%) | 24 (15.7%) | 0.96 | 32 (15.5%) | 48 (16%) | 0.888 | 50 (17.9%) | 30 (13.2%) | 0.149 |
| Diarrhoea | 88 (17.4%) | 66 (18.7%) | 22 (14.4%) | 0.239 | 47 (22.8%) | 41 (13.7%) | 0.008 | 55 (19.7%) | 33 (14.5%) | 0.127 |
| Nausea | 97 (19.2%) | 73 (20.7%) | 24 (15.7%) | 0.19 | 53 (25.7%) | 44 (14.7%) | 0.002 | 65 (23.3%) | 32 (14.1%) | 0.009 |
| Headache | 128 (25.3%) | 105 (29.8%) | 23 (15.0%) | <0.0001 | 69 (33.5%) | 59 (19.7%) | <0.0001 | 88 (31.5%) | 40 (17.6%) | <0.0001 |
| Dysgeusia | 117 (23.1%) | 102 (28.9%) | 15 (9.8%) | <0.0001 | 66 (32.0%) | 51 (17%) | <0.0001 | 86 (30.8%) | 31 (13.7%) | <0.0001 |
| Anosmia | 99 (19.6%) | 85 (24.1%) | 14 (9.2%) | <0.0001 | 53 (25.7%) | 46 (15.3%) | 0.004 | 68 (24.4%) | 31 (13.7%) | 0.003 |
| Fever | 231 (45.7%) | 166 (47.0%) | 65 (42.5%) | 0.346 | 99 (48.1%) | 132 (44.0%) | 0.368 | 131 (46.6%) | 100 (44.1%) | 0.515 |
| **Support during presentation, N (%)** | | | | | | | | | | |
| Supplemental oxygen or intubation | 404 (82.5%) | 272 | 131 | <0.0001 | 147 | 256 | <0.0001 | 201 | 202 | <0.0001 |
| Vasopressors on arrival | 58 (12.4%) | 19 (5.8%) | 39 (27.5%) | <0.0001 | 2 (1.0%) | 56 (20.4%) | <0.0001 | 4 (1.5%) | 54 (25.8%) | <0.0001 |
| Vital Signs, mean ± SD | | | | | | | | | | |
| Systolic blood pressure | 124.4±23.2 | 126.4±21.1 | 119.9±26.9 | 0.0066 | 122.0±24.5 | 128.1±20.5 | 0.0072 | 119.5±25.1 | 128.7±20.4 | <0.0001 |
| Diastolic blood pressure | 75.1±15.1 | 76.4±13.7 | 72.2±17.8 | 0.0079 | 73.7±16.1 | 77.2±13.3 | 0.0174 | 72.1±16.9 | 77.7±12.9 | 0.0001 |
| Heart rate | 88.6±16.4 | 87.3±16.4 | 91.8±16.0 | 0.0062 | 90.0±16.2 | 86.5±16.5 | 0.0203 | 90.5±16.1 | 87.0±16.5 | 0.0208 |
| Respiratory rate | 26.0±7.0 | 26.0±6.8 | 26.0±7.5 | 0.942 | 26.6±7.3 | 25.1±6.4 | 0.0311 | 26.7±7.6 | 25.4±6.4 | 0.0702 |
| Blood O2 saturation | 92.5±4.9 | 92.9±4.4 | 91.5±6.0 | 0.0055 | 91.8±5.7 | 93.6±3.3 | 0.0001 | 91.5±6.0 | 93.4±3.7 | <0.0001 |
| Comorbidities, N (%) | | | | | | | | | | |
| None | 112 (22.1) | 88 (24.9%) | 24 (15.7%) | 0.0036 | 41 (19.9%) | 71 (23.7%) | 0.815 | 57 (20.4%) | 55 (24.2%) | 0.217 |
| 1 comorbidity | 145 (28.7%) | 104 (29.5%) | 41 (26.8%) | - | 62 (30.1%) | 83 (27.7%) | - | 78 (28.0%) | 67 (29.5%) | - |
| 2 comorbidities | 142 (28.1%) | 88 (24.9%) | 54 (35.3%) | - | 61 (29.6%) | 81 (27.0%) | - | 76 (27.2%) | 66 (29.1%) | - |
| 3 comorbidities | 69 (13.6%) | 50 (14.2%) | 19 (12.4%) | - | 28 (13.6%) | 41 (13.7%) | - | 47 (16.9%) | 22 (9.7%) | - |
| ≥4 comorbidities | 38 (7.5%) | 23 (6.5%) | 15 (9.8%) | - | 14 (6.8%) | 24 (8.0%) | - | 21 (7.5%) | 17 (7.5%) | - |
| Hypertension | 280 (55.3%) | 192 (54.4%) | 88 (57.5%) | 0.516 | 112 (54.4%) | 168 (56.0%) | 0.717 | 159 (57.0%) | 121 (53.3%) | 0.407 |
| Diabetes | 181 (35.8%) | 119 (33.7%) | 62 (40.5%) | 0.142 | 63 (30.6%) | 118 (39.3%) | 0.044 | 92 (33.0%) | 89 (39.2%) | 0.146 |
| Past or current smoker | 144 (39.2%) | 100 (36.5%) | 44 (47.3%) | 0.065 | 62 (36.2%) | 82 (41.8%) | 0.275 | 90 (38.5%) | 54 (40.6%) | 0.687 |

(*Continued*)

**Table 1.** (Continued)

| Variable | All patients | Survivors | Non-survivors | P | Non-ICU patients | ICU patients | p | Non-intubated patients | Intubated patients | p |
|---|---|---|---|---|---|---|---|---|---|---|
| **Renal replacement therapy** | 15 (3.0%) | 8 (2.3%) | 7 (4.6%) | 0.16 | 6 (2.9%) | 9 (3.0%) | 0.955 | 7 (2.5%) | 8 (3.5%) | 0.503 |
| **Congestive heart failure** | 39 (7.7%) | 25 (7.1%) | 14 (9.2%) | 0.423 | 15 (7.3%) | 24 (8.0%) | 0.766 | 21 (7.5%) | 18 (7.9%) | 0.866 |
| **COPD** | 15 (3.0%) | 5 (1.4%) | 11 (7.2%) | 0.002 | 3 (1.5%) | 12 (4.0%) | 0.097 | 7 (2.5%) | 8 (3.5%) | 0.503 |
| **Asthma** | 22 (4.4%) | 14 (4.0%) | 10 (6.5%) | 0.522 | 11 (5.3%) | 11 (3.7%) | 0.365 | 13 (4.7%) | 9 (4.0%) | 0.703 |
| **Other lung diseases** | 12 (2.4%) | 9 (2.6%) | 8 (5.2%) | 0.689 | 4 (1.9%) | 8 (2.7%) | 0.599 | 6 (2.2%) | 6 (2.6%) | 0.717 |
| **Cancer** | 39 (7.7%) | 18 (5.1%) | 21 (13.7%) | 0.001 | 20 (9.7%) | 19 (6.3%) | 0.162 | 25 (9.0%) | 14 (6.2%) | 0.241 |
| **Solid organ transplant** | 16 (3.2%) | 8 (2.3%) | 8 (5.2%) | 0.08 | 5 (2.4%) | 11 (3.7%) | 0.434 | 7 (2.5%) | 9 (4.0%) | 0.352 |
| **Systemic lupus erythematosus** | 5 (1.0%) | 5 (1.4%) | 0 (0%) | 0.139 | 3 (1.5%) | 2 (0.7%) | 0.378 | 4 (1.4%) | 1 (0.4%) | 0.261 |
| **HIV** | 7 (1.4%) | 4 (1.1%) | 3 (2.0%) | 0.464 | 2 (1.0%) | 5 (1.7%) | 0.51 | 3 (1.1%) | 4 (1.8%) | 0.511 |
| **Medications, N (%)** | | | | | | | | | | |
| **Chloroquine** | 28 (6.0%) | 18 (5.4%) | 10 (7.5%) | 0.39 | 5 (2.5%) | 23 (8.6%) | 0.007 | 9 (3.4%) | 19 (9.6%) | 0.005 |
| **ACEIs** | 87 (18.2%) | 70 (20.6%) | 17 (12.3%) | 0.034 | 41 (20.4%) | 46 (16.6%) | 0.289 | 60 (22.0%) | 27 (13.2%) | 0.014 |
| **ARBs** | 86 (18.2%) | 68 (20.1%) | 18 (13.3%) | 0.084 | 43 (21.6%) | 43 (15.7%) | 0.1 | 60 (22.1%) | 26 (12.9%) | 0.01 |
| **Immunosuppressors** | 21 (4.9%) | 14 (4.5%) | 7 (6.0%) | 0.51 | 9 (4.8%) | 12 (5.0%) | 0.919 | 13 (5.1%) | 8 (4.7%) | 0.859 |
| **NSAIDs** | 39 (9.1%) | 31 (9.9%) | 8 (6.9%) | 0.331 | 20 (10.6%) | 19 (7.9%) | 0.332 | 25 (9.7%) | 14 (8.2%) | 0.588 |

(5.4%) needed haemodialysis. All-cause mortality in patients not admitted to the ICU was 18 patients (8.7%), and in patients who did not need intubation, it was 26 (9.3%). In patients admitted to the ICU, 135 patients (45%) died, while 127 (56%) died in the intubated patients. Of all the patients on vasopressors, 114 (63.7%) died. Of all the patients on dialysis, 49 (72,1%) died. The median time from illness onset to ED evaluation was 8 days, and that from onset to home discharge was 22.1 days. The median length of hospital stay was 11 days (Fig 2).

## Comparison between survivors and non-survivors

Survivors were significantly younger than non-survivors (57.4 vs 66.2 years old). Myalgia (159 patients [45.0%] vs 38 [24.8%]), sore throat (60 [17.0%] vs 15 [9.8%]), headache (105 [29.8%] vs 23 [15.0%]), dysgeusia (102 [28.9%] vs 15 [9.8%]), and anosmia (85 [24.1%] vs 14 [9.2%]) were significantly more frequent among survivors. In our univariate analysis low blood pressure, high cardiac rate, and hypoxia were associated with an increased risk of death, ICU admission, or invasive ventilation (Table 1).

There were significantly higher levels of C-reactive protein, lactate dehydrogenase, neutrophils, and troponin T and lower levels of lymphocytes in: non-survivors than that in survivors, in patients admitted to the ICU than that in those not admitted to the ICU, and in intubated patients than that in those who were not intubated.

In this study, the D-dimer levels were high, with 332 (86.7%) patients having levels higher than 500 μg/dL, and with survivors having a median level of 1136 U/L, compared with 1851 U/L in non-survivors. Admission troponin T was available for 314 patients, and in 157 (50%) it was elevated. Median cardiac troponin T for survivors was 11 ng/L, while it was 38 ng/L for non-survivors. In survivors, 83 (38.3%) had an elevated troponin T level, while 74 (76.3%) had an elevated troponin T level in non-survivors.

Patients who were taking angiotensin-converting enzyme inhibitors (ACEIs) or angiotensin II type 1 receptor blockers (ARBs) at home had lower odds ratio for mortality. While chloroquine was not prescribed for COVID-19 treatment in our institution, 28 patients who arrived in our ED were already using that. Of those, we continued chloroquine in six

**Table 2. Laboratory test results of patients with COVID-19 pneumonia.**

| Variable | All patients | Survivors | Non-survivors | P | Non-ICU patients | ICU patients | p | Non-intubated patients | Intubated patients | p |
|---|---|---|---|---|---|---|---|---|---|---|
| Hb (n = 506) (g/dL) | 12.4±2.2 | 12.7±1.9 | 11.9±2.6 | 0.0002 | 12.3±2.3 | 12.5±2.1 | 0.4973 | 12.5±2.2 | 12.4±2.2 | 0.5325 |
| Neutrophils (n = 506) (cells/m$^3$ × 10$^3$) | 6850±4200 | 6200±3400 | 8400±5200 | <0.0001 | 5200±2900 | 7900±4500 | <0.0001 | 5600±3000 | 8300±4700 | <0.0001 |
| Lymphocytes (n = 430) (cells/m$^3$ × 10$^3$) | 1001±540 | 1077±553 | 825±464 | <0.0001 | 1130±532 | 921±530 | 0.0001 | 1093±514 | 900±551 | 0.0002 |
| Lymphopenia (<1500) | 356 (82.8%) | 242 (80.1%) | 114 (89.1%) | 0.025 | 125 (75.3%) | 231 (87.5%) | 0.001 | 176 (77.9%) | 180 (88.2%) | 0.004 |
| Platelets (n = 468) (cells/m$^3$ × 10$^3$) | 219,000 ±87,000 | 223,000 ±86,000 | 208,000 ±90,000 | 0.0947 | 221,000 ±86,000 | 217,000 ±88,000 | 0.6331 | 219,000±84,000 | 218,000 ±91,000 | 0.9538 |
| Creatinine (n = 470) (mg/dL) | 1.5±1.7 | 1.2±1.3 | 2.3±2.3 | <0.0001 | 1.2±1.5 | 1.7±1.8 | 0.0024 | 1.2±1.3 | 1.9±2.0 | <0.0001 |
| Creatinine (>1.2 mg/dL) | 160 (34%) | 81 (24.7%) | 79 (55.2%) | <0.0001 | 40 (21.7%) | 120 (42.0%) | <0.0001 | 56 (22.5%) | 104 (47.1%) | <0.0001 |
| D-dimer (n = 383) (U/L) | 1278 (676–2569)* | 1136 (630–2057)* | 1851 (850–4676)* | <0.0001 | 963 (580–1569)* | 1467 (750–3948)* | <0.0001 | 977 (565–1554)* | 1694 (835–4284)* | <0.0001 |
| Elevated D-dimer (>500 U/L) | 332 (86.7%) | 232 (83.5%) | 100 (95.2%) | 0.002 | 122 (80.8%) | 210 (90.5%) | 0.006 | 162 (80.2%) | 170 (93.9%) | 0.0001 |
| C-reactive protein (n = 436) (mg/dL) | 171±111 | 154±102 | 210±120 | <0.0001 | 115±84 | 207±111 | <0.0001 | 124±83 | 222±115 | <0.0001 |
| Elevated C-reactive protein (>100 mg/dL) | 299 (68.6%) | 194 (63.4%) | 105 (80.8%) | <0.0001 | 80 (46.5%) | 219 (83.0%) | <0.0001 | 123 (53.5%) | 176 (85.4%) | <0.0001 |
| AST (n = 418) (U/L) | 42 (30–61)* | 39 (28–58)* | 50 (34–74)* | 0.0005 | 36 (27–55) | 44 (33–67)* | 0.0004 | 38 (27–55)* | 50 (33–72)* | 0.0001 |
| AST (>40 U/L) | 219 (52.4%) | 138 (47.6%) | 81 (63.3%) | 0.003 | 73 (44.8%) | 146 (57.3%) | 0.013 | 96 (44.4%) | 123 (60.9%) | 0.001 |
| LDH (n = 373) (U/L) | 467±417 | 417±246 | 586±399 | <0.0001 | 375±271 | 529±317 | <0.0001 | 387±251 | 562±341 | <0.0001 |
| LDH (>250 U/L) | 337 (90.3%) | 232 (88.2%) | 105 (95.5%) | 0.031 | 126 (84.0%) | 211 (94.6%) | 0.001 | 174 (86.6%) | 163 (94.8%) | 0.008 |
| CPK (U/L) | 854±3669 | 680±3908 | 1260±3024 | 0.2623 | 380±1865 | 1118±4347 | 0.1354 | 315±1584 | 1392±4893 | 0.0227 |
| CPK (>200 U/L) | 82 (34.2%) | 46 (27.4%) | 36 (50.0%) | 0.001 | 20 (23.3%) | 62 (40.3%) | 0.008 | 25 (20.8%) | 57 (47.5%) | <0.0001 |
| Troponin T (n = 314) (ng/L) | 15 (8–45)* | 11 (7–20)* | 38 (15–96)* | 0.0001 | 11 (7–23)* | 17 (9–56)* | 0.0009 | 11 (7–22)* | 21 (10–83)* | 0.0001 |
| Elevated troponin | 157 (50.0%) | 83 (38.3%) | 74 (76.3%) | <0.0001 | 47 (40.2%) | 110 (55.8%) | 0.007 | 61 (38.6%) | 96 (61.5%) | <0.0001 |

Number in parenthesis indicates number of patients who had that test ordered in the emergency department.

* Median and interquartile-range. COVID-19, coronavirus disease;, ICU, intensive care unit; Hb, haemoglobin; AST, aspartate aminotransferase; LDH, lactate dehydrogenase; CPK, creatine phosphokinase.

patients who were taking it or hydroxychloroquine at home, while we suspended the medication for the others. Although we observed that chloroquine or hydroxychloroquine was associated with increased mortality, but because the number of patients was only 28, we could not draw any meaningful conclusion.

Table 3 shows that of the 300 patients admitted to the ICU, 135 (45.0%) died, 146 (48.7%) were discharged, and 19 (6.3%) remained in the hospital. Meanwhile, of the 227 patients who needed invasive ventilation, 127 (55.9%) died, 82 (36.1%) were discharged home, and 18 (7.9%) remained in the hospital.

The mortality rate in patients who needed vasopressors and dialysis support was 63.7% (114 deaths out of 179 patients on vasopressors) and 72.1% (49 deaths out of 68 patiens on haemodialysis), respectively. Fifty-two (29.1%) of the patients who needed vasopressors were discharged home, while only 12 of those who needed dialysis were discharged home (17.6%).

A multivariate analysis revealed that age, number of comorbidities, increased extension of ground glass on CT, increased troponin T were associated with all-cause mortality whereas use of angiotensin converting enzyme inhibitor, use of angiotensin II receptor blocker, higher

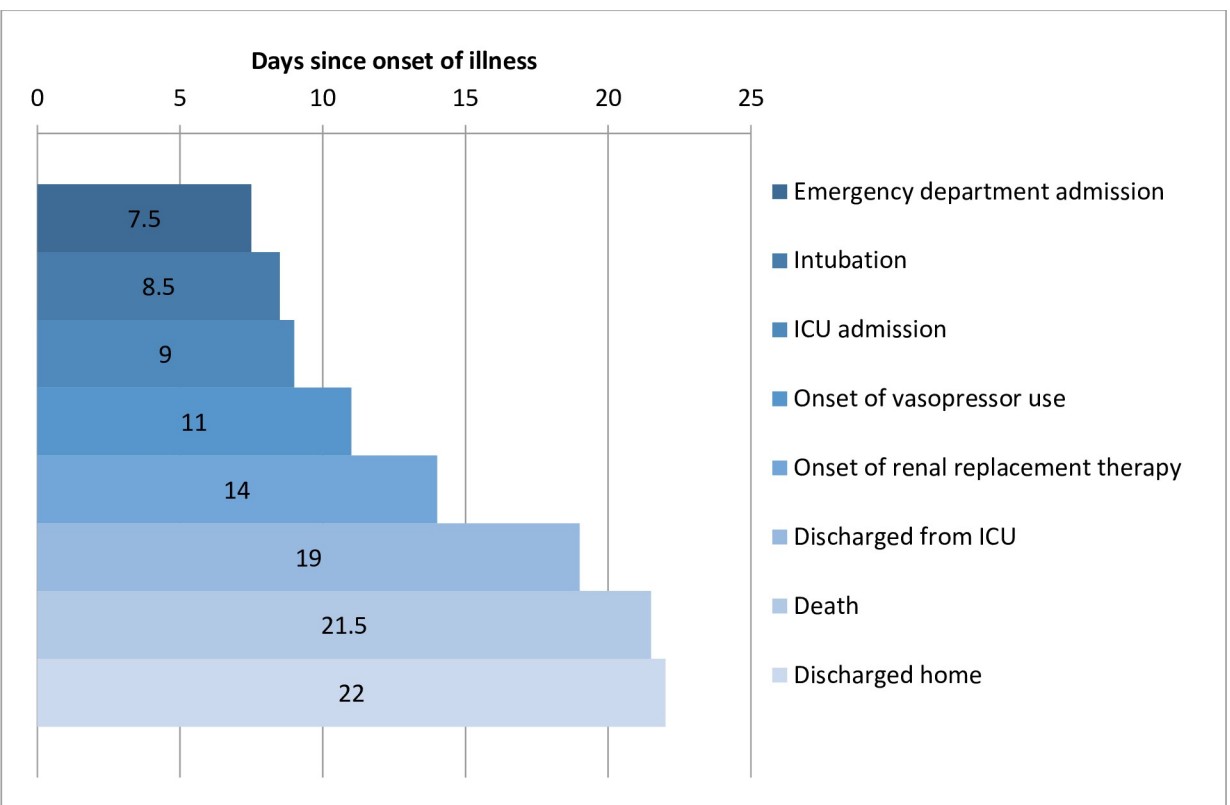

**Fig 2. Timeline of coronavirus disease (COVID-19) cases after onset of illness.** *: ICU intensive care unit.

number of lymphocytes, and dysgeusia were associated with decreased all-cause mortality (Table 4).

We applied our final model back on our population to gage the multivariate model. We found a sensitivity of 81%, specificity of 71% for all-cause mortality. The negative predictive value was 87% while the positive predictive value was 62% and the AUC was 0.82 (0.78–0.86).

## Discussion

This prospective study evaluated a cohort of patients who presented to the ED with severe COVID-19. Among the patients, 59.2% were admitted to the ICU and 41.5% needed

**Table 3. Mechanical ventilation, ICU admission and all-cause mortality of patients with COVID-19 pneumonia.**

| Outcomes | All patients | Survivors | Non-survivors | ICU patients | Non-ICU patients | Intubated patients | Non intubated | Vasopressors | Dialytic patients |
|---|---|---|---|---|---|---|---|---|---|
| N | 506 | 353 | 153 | 300 | 206 | 227 | 279 | 179 | 68 |
| **Orotracheal intubation** | 227 (44.9%) | 100 (28.3%) | 127 (83.0%) | 226 (75.3%) | – | – | – | 170 (94.5%) | 65 (98.5%) |
| **ICU** | 300 (59.3%) | 165 (46.7%) | 135 (88.2%) | – | – | 226 (99.6%) | 74 (26.5%) | 179 (100%) | 68 (100%) |
| **Vasopressor** | 179 (35.4%) | 65 (18.4%) | 114 (74.5%) | 179 (59.7%) | – | 170 (74.9%) | 9 (3.2%) | – | 66 (97.1%) |
| **Haemodialysis** | 68 (13.4%) | 19 (5.4%) | 49 (32.0%) | 68 (22.7%) | – | 67 (29.5%) | 1 (0.4%) | 66 (36.9%) | – |
| **All-cause mortality** | 153 (30.2%) | – | – | 135 (45.0%) | 18 (8.7%) | 127 (55.6%) | 26 (9.3%) | 114 (63.7%) | 49 (72.1%) |

**Table 4. Multivariate predictors of the risk of death.**

| Multivariate predictor | Beta coefficient |
|---|---|
| Intercept | -3.406 |
| Age | .036 |
| Dysgeusia | -0.801 |
| Comorbidities | 0.253 |
| Use of ACEIs | -1.332 |
| Use of ARBs | -1.104 |
| Lymphocytes | -0.802 |
| Troponin T | 3.062 |
| Extension of ground glass on CT | 0.381 |

CI, confidence interval; CT, computed tomography; ACEIs, angiotensin-converting enzyme inhibitors, ARBs, angiotensin II type 1 receptor blockers

* lambda of 0,0007498653 and deviance of 0,9156566

mechanical ventilator support, far greater than the corresponding percentages in other series. Unlike other published cohorts, the vast majority (96%) of patients in this study have already had a defined outcome [16–19].

In the largest ICU series of cases from Lombardy in Italy, 58.2% of patients remained in the ICU. When only patients with defined outcomes were evaluated, the mortality in the ICU was 62.2% [20] Data from the Brazilian Society of Intensive Care shows that ICU mortality in public hospitals is 51.7% and 29.1% in private hospitals [21]. In our study we found 45.0% ICU mortality, which is better than other series, but not as good as that recorded for private hospitals in this country. Moreover, the mortality of intubated patients in our study was high, at 55.9%, while only 36.1% of these patients were discharged. This mortality rate is still lower than that of some of the largest series of cases, such as the New York City series, which had a mortality rate of 88.8% for intubated patients [18]. Other series showed mortality rates of intubated patients as high as 97% [22–24]. Another study however, found a considerably lower mortality rate of approximately 35% in intubated patients [25].

In our univariate analysis, we found that levels of cardiac troponin T, D-dimer, and changes to other laboratory markers were associated with mortality, mechanical ventilation, and ICU admission. There were 55 patients with D-dimer levels over 5000 U/L, ranging from 5260 to 107,138 U/L. Zhou and colleagues found that D-dimer levels were higher than 1000 μg/dL in 82% of non-survivors, compared with 24% of survivors, in a case series of COVID-19 pneumonia [26]. Troponin increase likely reflects increased acute coronary syndrome secondary to infectious state but also myocardial injury secondary to COVID-19-related prothrombotic and proinflammatory states [27].

The LASSO multivariate analysis found age, number of comorbidities, extension of ground glass opacities on chest CT and troponin with a direct relationship with all-cause mortality, whereas dysgeusia, use of angiotensin converting enzyme inhibitor or angiotensin-ii receptor blocker and number of lymphocytes with an inverse relationship with all-cause mortality In different studies older age, hypertension, diabetes melitus, dyspnea, number of comorbidities, and laboratory parameters were associated to increased risk of mortality in patients with COVID-19 [28, 29]. The symptoms at presentation in our study were not significantly different compared with those of other large studies [16, 30], even considering that our patients had, on average, more severe disease. This finding is unsurprising because most of the symptoms were not significantly different among survivors and non-survivors except for myalgia,

headache, anosmia, and dysgeusia associated with lower mortality in an univariate analysis, while dysgeusia remained significant in the multivariate analysis. To our knowledge, this is the first study that found such association. In a cohort of 345 patients, of whom a high proportion had olfactory and gustatory function impairment, there were no significant differences in the severity of disease and outcomes in patients with such impairment [31].

COVID-19 may have a progressive evolution. Huang and colleagues reported that patients with COVID-19 were admitted to the ICU after 11 days of symptoms [1, 26] and intubated after 12 days of symptoms. In our cohort, this timeline was a little shorter–our patients were admitted to the ICU and were intubated in a median of 8.7 days and 8.9 days, respectively, after the onset of symptoms.

Interestingly, patients who were taking ACEIs or ARBs at home had a reduced mortality. ACEIs or ARBs do not increase the risk of COVID-19 [32]. Some studies have suggested a potential beneficial effect of RAS inhibitors in SARS-CoV-2 patients, and this has been put forward in some reviews [33–38]. SARS-CoV-2 enters type 2 pneumocytes in humans through angiotensin-converting enzyme 2 receptors [39]. Several studies, including our own, showed hypertension as the most common comorbidity associated with COVID-19 [16, 18, 30]. In this study, typical patients with severe COVID-19 had their antihypertensives suspended on admission and reintroduced upon stabilisation of blood arterial pressure, usually on discharge to the ward. New studies are necessary to test the association of ACEIs and ARBs with COVID-19 outcomes. The PRAETORIAN-COVID trial that is a double-blind, placebo-controlled randomised clinical trial of valsartan to prevent acute respiratory distress syndrome in hospitalised patients with SARS-COV-2 infection, may help to solve this question [40]. The BRACE CORONA trial evaluated temporary interruption of ACEi/ARB at COVID-19 diagnosis. They evaluated a low risk cohort–they reported general all-cause mortality was 2.74%. They found no difference in number of days alive and out of hospital through 30 days or all-cause mortality. A hypothesis that could be made is that the ACEi/ARB benefit we observed is because of the severity of our cohort [41]. The benefit we observed could be a statistical fluke and would need to be confirmed in other cohorts for validaiton.

Our study has some limitations. It was a single-centre study, and patients from other EDs were referred to our ED. Furthermore, our patients had COVID-19 that was more severe than what would be ordinarily expected in the ED [16–19].

In conclusion, this is a case series of older patients with multiple comorbidities presenting with moderate and severe COVID-19 pneumonia in São Paulo city, Brazil. The majority were admitted to the ICU and many were intubated, treated with vasopressors, and haemodialysis. All-cause mortality was 30% for the entire series, 45% for patients admitted to the ICU and 56% for the intubated patients. Finally, we tested the model in the same database from which the model was derived.

Preparation to adequately treat patients with severe COVID-19 involves providing not only large numbers of ICU beds but also warrants a high capacity to provide mechanical ventilation, vasopressors, and haemodialysis.

## Supporting information

**S1 Data.**
(XLS)

## Acknowledgments

The Emergencia USP Covid Group is composed by

Felipe Liger Moreira, MD**1, Edwin Albert D'Souza*1, Arthur Petrillo Bellintani*1, Rodrigo Cezar Miléo*1, Rodrigo Werner Toccoli*1, Fernanda Máximo Fonseca e Silva*1, João Martelleto Baptista*1, Marcelo de Oliveira Silva*1, Giovanna Babikian Costa*1, Rafael Berenguer Luna*1, Henrique Tibucheski dos Santos*1, Mariana Mendes Gonçalves Cimatti De Calasans*1, Marcelo Petrof Sanches*1, Diego Juniti Takamune*1, Luiza Boscolo*1, Pedro Antonio Araújo Simões*1, Manuela Cristina Adsuara Pandolfi*1, Beatriz Larios Fantinatti*1, Gabriel Travessini*1, Matheus Finardi Lima de Faria*1, Ligia Trombetta Lima*1, Bianca Ruiz Nicolao*1, Gabriel de Paula Maroni Escudeiro*1, João Pedro Afonso Nascimento*1, Everton Luis Santos Moreira*1, Erika Thiemy Brito Miyaguchi *1, Bruna Tolentino Caldeira*1, Laura de Góes Campos*1, Vitor Macedo Brito Medeiros*1, Tales Cabral Monsalvarga*1, Isabela Harumi Omori*1, Diogo Visconti Guidotte*1, Alexandre Lemos Bortolotto*1, Rodrigo de Souza Abreu*1, Nilo Arthur Bezerra Martins*1, Carlos Eduardo Umehara Juck*1, Natalia Paula Cardos*1, Osvaldo Santistevan Claure*1, João Vitor Ziroldo Lopes*1, Felipe Mouzo Bortoleto**, MD1, Gabriel Martinez**, MD1, Lucas Gonçalves Dias Barreto**, MD, Debora Lopes Emerenciano**, MD1, Daniel Rodrigues Ribeiro**, MD1, Danilo Dias de Francesco**, MD1, Eduardo Mariani Pires de Campos**, MD1, Stefany Franhan Barbosa de Souza**, MD1, Geovane Wiebelling da Silva**, MD1, Andrew Araujo Tavares**, MD1, Clara Carvalho de Alves Pereira**, MD1, Ademar Lima Simões**, MD1, Gustavo Biz Martins**, MD1, Maria Lorraine Silva de Rosa**, MD1, Thiago Areas Lisboa Netto**, MD1, Julio Cesar Leite Fortes**, MD1, Rafael Faria Pisciolaro**, MD1, Mauricio Ursoline do Nascimento**, MD1, Rodolfo Affonso Xavier**, MD1, Yago Henrique Padovan Chio**, MD1, Patricia Albuquerque de Moura***, BS1, Emily Cristine Oliveira Silva***, BS1, Ester Minã Gomes da Silva***, BS1, Yasmine Souza Filippo Fernandes***, BS1, Renata Kan Nishiaka***, BS1,

* Medical Student
** Emergency medicine resident
*** Respiratory physiotherapy specialist
1Emergency Department, Hospital das Clínicas da Faculdade de Medicina da Universidade de São Paulo

## Author Contributions

**Conceptualization:** Rodrigo A. Brandão Neto, Julio F. Marchini, Lucas O. Marino.

**Data curation:** Rodrigo A. Brandão Neto, Lucas O. Marino, Julio C. G. Alencar, Sabrina Ribeiro, Caue G. Bueno, Victor P. da Cunha, Heraldo Possolo de Souza.

**Formal analysis:** Rodrigo A. Brandão Neto, Julio F. Marchini, Luz Marina Gomez Gomez, Heraldo Possolo de Souza.

**Investigation:** Lucas O. Marino, Felippe Lazar Neto, Sabrina Ribeiro, Fernando V. Salvetti, Hassan Rahhal, Caue G. Bueno, Carine C. Faria, Victor P. da Cunha, Eduardo Padrão.

**Methodology:** Julio F. Marchini, Lucas O. Marino, Felippe Lazar Neto, Heraldo Possolo de Souza.

**Project administration:** Rodrigo A. Brandão Neto, Julio F. Marchini, Julio C. G. Alencar, Caue G. Bueno, Irineu T. Velasco, Heraldo Possolo de Souza.

**Resources:** Julio C. G. Alencar, Caue G. Bueno.

**Software:** Julio F. Marchini, Felippe Lazar Neto.

**Supervision:** Rodrigo A. Brandão Neto, Julio F. Marchini, Irineu T. Velasco, Heraldo Possolo de Souza.

**Validation:** Caue G. Bueno.

**Writing – original draft:** Rodrigo A. Brandão Neto, Heraldo Possolo de Souza.

**Writing – review & editing:** Rodrigo A. Brandão Neto, Julio F. Marchini, Lucas O. Marino, Sabrina Ribeiro, Fernando V. Salvetti, Hassan Rahhal, Luz Marina Gomez Gomez, Irineu T. Velasco, Heraldo Possolo de Souza.

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
