## [Decision Letter · Decision Letter 0]

27 Oct 2020

PONE-D-20-29947

Mortality and other outcomes of patients with coronavirus disease pneumonia admitted to the emergency department: a prospective observational Brazilian study.

PLOS ONE

Dear Dr. Marchini,

Thank you for submitting your manuscript to PLOS ONE. After careful consideration, we feel that it has merit but does not fully meet PLOS ONE’s publication criteria as it currently stands. Therefore, we invite you to submit a revised version of the manuscript that addresses the points raised during the review process.

We look forward to receiving your revised manuscript.

Kind regards,

Walter R. Taylor

Academic Editor

PLOS ONE

Journal Requirements:

2. In the Methods, please provide further details about your consent. Please confirm:

- Whether the Institutional Review Board approved the use of verbal authorization from the patient or family was obtained in the presence of witnesses

- How verbal authorization from the patient or family was documented

For more information, please see our guidelines for human subjects research: https://journals.plos.org/plosone/s/submission-guidelines#loc-human-subjects-research

3. One of the noted authors is a group; Emergencia USP Covid group.

In addition to naming the author group, please list the individual authors and affiliations within this group in the acknowledgments section of your manuscript.

Please also indicate clearly a lead author for this group along with a contact email address.

4. We note you have included a table to which you do not refer in the text of your manuscript. Please ensure that you refer to Table 3 in your text; if accepted, production will need this reference to link the reader to the Table.

5. Please include captions for your Supporting Information files at the end of your manuscript, and update any in-text citations to match accordingly. Please see our Supporting Information guidelines for more information: http://journals.plos.org/plosone/s/supporting-information

Additional Editor Comments:

Dear Dr. Marchini,

thankyou for your nice paper.

I have reviewed it and received comments from one excellent reviewer who makes very useful and constructive comments.

yours sincerely,

Walter Taylor

Reviewers' comments:

Reviewer's Responses to Questions

**Comments to the Author**

1. Is the manuscript technically sound, and do the data support the conclusions?

Reviewer #1: Yes

2. Has the statistical analysis been performed appropriately and rigorously? 

Reviewer #1: I Don't Know

3. Have the authors made all data underlying the findings in their manuscript fully available?

Reviewer #1: Yes

4. Is the manuscript presented in an intelligible fashion and written in standard English?

Reviewer #1: Yes

5. Review Comments to the Author

Reviewer #1: Thank you for the opportunity to review this paper. It is good to read a report describing presentations with COVID-19 that does not come from one of the countries that have tended to dominate the literature on this topic. Overall, the study provides a useful addition to the emerging literature on acute presentations with COVID-19. I have the following suggestions that I hope the authors will find helpful.

Please state the aim and specific objectives of the study at the end of the introduction, ensuring that the objectives are in accordance with the analysis undertaken (i.e. to describe the characteristics of people admitted to hospital with COVID-19 pneumonia and identify independent predictors of adverse outcome).

Under study design, it would be helpful to know the population covered by the hospital. Did the hospital act as the referral centre for the whole of Sao Paulo and, if so, what is the population of Sao Paulo?

Non-invasive ventilation is not reported in the outcomes. There have been reports that non-invasive ventilation (including CPAP) can reduce the need for ventilation and suggestions that this may improve outcomes. Are you able to comment on the use of non-invasive ventilation in your hospital?

Please clarify the duration of follow up? Presumably patients were follow until death or hospital discharge, but a proportion were still in hospital at a specified date.

How were the 2219 ED attendances selected? I presume these were patients with suspected COVID-19, so how was that defined? I also assume that only those admitted were routinely tested for COVID-19, but please clarify this.

It would be interesting to compare the characteristics and outcomes of those admitted with confirmed COVID-19 to those with negative COVID-19 testing. Maybe this is planned for a separate analysis.

I recommend reporting numbers alongside the percentages in the results text.

The description of the multivariable analysis requires some more detail to allow it to be reproducible:

1. How were missing predictor variables handled (I assume those with missing outcomes were excluded from the analysis)? Were missing data assumed to be normal? This may require some more information regarding the recording of predictor variables, perhaps by providing the data collection form as an appendix.

2. How was the relationship between continuous predictor variables and outcome modelled, given that some may have non-linear relationships with outcome?

3. How many predictor variables were included in the analysis? The sample size may be insufficient for a large number of predictor variables, and including too many may lead to over-fitting.

4. Was any analysis undertaken to validate the multivariable model?

In general, I think the multivariable analysis is the weakest part of the study. The descriptive analysis is appropriate and clearly presented, allowing readers to understand the characteristics of your population and compare it to other populations. It is more difficult to draw conclusions from the multivariable analysis. The study may lack power for multivariable analysis, creating risks of important predictors being missed, while the model may be over-fitted if too many predictor variables were included. I would either drop the multivariable analysis from the paper or, if retained, ensure the analysis is clearly described and the limitations acknowledged.

Finally, PLOS ONE has reviewed a paper from my research team describing ED attendances with suspected COVID-19 across the UK. You may wish to contrast your findings with ours in your discussion. The pre-print is available here (https://www.medrxiv.org/content/10.1101/2020.08.10.20171496v1) and I am happy to share our revised paper, if this is OK with the PLOS ONE editors.

Steve Goodacre, 12 October 2020

6. PLOS authors have the option to publish the peer review history of their article (what does this mean?). If published, this will include your full peer review and any attached files.

Reviewer #1: **Yes: **Steve Goodacre

---

## [Author Response · Author response to Decision Letter 0]

4 Dec 2020

Comments from the editor:

Comment #1: Whether the Institutional Review Board approved the use of verbal authorization from the patient or family was obtained in the presence of witnesses. How verbal authorization from the patient or family was documented.

Response #1: In document #4.1119.252 from the HCFMUSP research ethics committee approved that verbal authorization can be obtained in the presence of witnessess and this should be documented in the patient's chart.

Comment #2: One of the noted authors is a group; Emergencia USP Covid group.

In addition to naming the author group, please list the individual authors and affiliations within this group in the acknowledgments section of your manuscript.

Please also indicate clearly a lead author for this group along with a contact email address.

Response #2: We have included this infomartion in the acknowledgements section and included more information on their affiliations 

Original version:

Emergencia USP Covid Group

Felipe Liger Moreira, MD1 , Edwin Albert D’Souza*1 , Arthur Petrillo Bellintani*1 , Rodrigo Cezar Miléo*1 , Rodrigo Werner Toccoli*1, Fernanda Máximo Fonseca e Silva*1, João Martelleto Baptista*1 , Marcelo de Oliveira Silva*1 , Giovanna Babikian Costa*1 , Rafael Berenguer Luna*1 , Henrique Tibucheski dos Santos*1 , Mariana Mendes Gonçalves Cimatti De Calasans*1, Marcelo Petrof Sanches*1 , Diego Juniti Takamune*1 , Luiza Boscolo*1 , Pedro Antonio Araújo Simões*1 , Manuela Cristina Adsuara Pandolfi*1 , Beatriz Larios Fantinatti*1 , Gabriel Travessini*1 , Matheus Finardi Lima de Faria*1 , Ligia Trombetta Lima*1 , Bianca Ruiz Nicolao*1 , Gabriel de Paula Maroni Escudeiro*1 , João Pedro Afonso Nascimento*1 , Everton Luis Santos Moreira*1 , Erika Thiemy Brito Miyaguchi *1 , Bruna Tolentino Caldeira*1 , Laura de Góes Campos*1 , Vitor Macedo Brito Medeiros*1 , Tales Cabral Monsalvarga*1 , Isabela Harumi Omori*1 , Diogo Visconti Guidotte*1 , Alexandre Lemos Bortolotto*1 , Rodrigo de Souza Abreu*1 , Nilo Arthur Bezerra Martins*1 , Carlos Eduardo Umehara Juck*1 , Felipe Mouzo Bortoleto, MD1 , Gabriel Martinez, MD1 , Lucas Gonçalves Dias Barreto, MD1 , Debora Lopes Emerenciano, MD1 , Daniel Rodrigues Ribeiro, MD1 , Danilo Dias de Francesco, MD1 , Eduardo Mariani Pires de Campos, MD1 , Stefany Franhan Barbosa de Souza, MD1 , Geovane Wiebelling da Silva, MD1 , Andrew Araujo Tavares, MD1 , Clara Carvalho de Alves Pereira, MD1 , Ademar Lima Simões, MD1 , Gustavo Biz Martins, MD1 , Maria Lorraine Silva de Rosa, MD1 , Thiago Areas Lisboa Netto, MD1 , Julio Cesar Leite Fortes, MD1 , Rafael Faria Pisciolaro, MD1 , Mauricio Ursoline do Nascimento, MD1 , Rodolfo Affonso Xavier, MD1 , Yago Henrique Padovan Chio, MD1 , Patricia Albuquerque de Moura, BS1 , Emily Cristine Oliveira Silva, BS1 , Ester Minã Gomes da Silva, BS1 , Yasmine Souza Filippo Fernandes, BS1 , Renata Kan Nishiaka, BS1 , 

* Medical Student 

1Emergency Department, Hospital das Clínicas da Faculdade de Medicina da Universidade de São Paulo 

Revised version:

Acknowledgements 

The Emergencia USP Covid Group is composed by

Felipe Liger Moreira, MD**1 , Edwin Albert D’Souza*1 , Arthur Petrillo Bellintani*1 , Rodrigo Cezar Miléo*1 , Rodrigo Werner Toccoli*1, Fernanda Máximo Fonseca e Silva*1, João Martelleto Baptista*1 , Marcelo de Oliveira Silva*1 , Giovanna Babikian Costa*1 , Rafael Berenguer Luna*1 , Henrique Tibucheski dos Santos*1 , Mariana Mendes Gonçalves Cimatti De Calasans*1, Marcelo Petrof Sanches*1 , Diego Juniti Takamune*1 , Luiza Boscolo*1 , Pedro Antonio Araújo Simões*1 , Manuela Cristina Adsuara Pandolfi*1 , Beatriz Larios Fantinatti*1 , Gabriel Travessini*1 , Matheus Finardi Lima de Faria*1 , Ligia Trombetta Lima*1 , Bianca Ruiz Nicolao*1 , Gabriel de Paula Maroni Escudeiro*1 , João Pedro Afonso Nascimento*1 , Everton Luis Santos Moreira*1 , Erika Thiemy Brito Miyaguchi *1 , Bruna Tolentino Caldeira*1 , Laura de Góes Campos*1 , Vitor Macedo Brito Medeiros*1 , Tales Cabral Monsalvarga*1 , Isabela Harumi Omori*1 , Diogo Visconti Guidotte*1 , Alexandre Lemos Bortolotto*1 , Rodrigo de Souza Abreu*1 , Nilo Arthur Bezerra Martins*1 , Carlos Eduardo Umehara Juck*1 , Felipe Mouzo Bortoleto**, MD1 , Gabriel Martinez**, MD1 , Lucas Gonçalves Dias Barreto**, MD1 , Debora Lopes Emerenciano**, MD1 , Daniel Rodrigues Ribeiro**, MD1 , Danilo Dias de Francesco**, MD1 , Eduardo Mariani Pires de Campos**, MD1 , Stefany Franhan Barbosa de Souza**, MD1 , Geovane Wiebelling da Silva**, MD1 , Andrew Araujo Tavares**, MD1 , Clara Carvalho de Alves Pereira**, MD1 , Ademar Lima Simões**, MD1 , Gustavo Biz Martins**, MD1 , Maria Lorraine Silva de Rosa**, MD1 , Thiago Areas Lisboa Netto**, MD1 , Julio Cesar Leite Fortes**, MD1 , Rafael Faria Pisciolaro**, MD1 , Mauricio Ursoline do Nascimento**, MD1 , Rodolfo Affonso Xavier**, MD1 , Yago Henrique Padovan Chio**, MD1 , Patricia Albuquerque de Moura***, BS1 , Emily Cristine Oliveira Silva***, BS1 , Ester Minã Gomes da Silva***, BS1 , Yasmine Souza Filippo Fernandes***, BS1 , Renata Kan Nishiaka***, BS1 , 

* Medical Student 

** Emergency medicine resident

*** Respiratory physiotherapy specialist

1Emergency Department, Hospital das Clínicas da Faculdade de Medicina da Universidade de São Paulo 

Comment #3: We note you have included a table to which you do not refer in the text of your manuscript. Please ensure that you refer to Table 3 in your text; if accepted, production will need this reference to link the reader to the Table.

Response #3: Thank for pointing out this out. We are now referencing Table 3 in the text.

Original text:

Of the 300 patients admitted to the ICU, 135 (45.0%) died, 146 (48.7%) were discharged, and 19 (6.3%) remained in the hospital. 

Revised text:

Table 3 shows that of the 300 patients admitted to the ICU, 135 (45.0%) died, 146 (48.7%) were discharged, and 19 (6.3%) remained in the hospital.

Comment #4: Please include captions for your Supporting Information files at the end of your manuscript, and update any in-text citations to match accordingly. Please see our Supporting Information guidelines for more information: http://journals.plos.org/plosone/s/supporting-information

Response #4: Our full database has been made available to PLOS ONE. I have translated any portuguese terms into english and reupdated the new version.

---

## [Editor Report · Decision Letter 1]

14 Dec 2020

Mortality and other outcomes of patients with coronavirus disease pneumonia admitted to the emergency department: a prospective observational Brazilian study.

PONE-D-20-29947R1

Dear Dr. Marchini,

We’re pleased to inform you that your manuscript has been judged scientifically suitable for publication and will be formally accepted for publication once it meets all outstanding technical requirements.

Kind regards,

Walter R. Taylor

Academic Editor

PLOS ONE

Additional Editor Comments (optional):

Dear Dr. Marchini,

thankyou for the revision of this paper.

I am happy to accept it for publication.

yours sincerely,

Walter Taylor.
---

## [Editor Report · Acceptance letter]

16 Dec 2020

PONE-D-20-29947R1 

Mortality and other outcomes of patients with coronavirus disease pneumonia admitted to the emergency department: a prospective observational Brazilian study 

Dear Dr. Marchini:

I'm pleased to inform you that your manuscript has been deemed suitable for publication in PLOS ONE. Congratulations! Your manuscript is now with our production department. 

Kind regards, 

on behalf of

Dr. Walter R. Taylor 

Academic Editor

PLOS ONE